# Interferon-based agents for current and future viral respiratory infections: A scoping literature review of human studies

Aldina Mesic[1,2]*, Emahlea K. Jackson[1,3], Mathias Lalika[1,2], David M. Koelle[2,4,5,6,7], Rena C. Patel[1,2,4]

**1** Department of Global Health, The Strategic Analysis, Research & Training (START) Center, University of Washington, Seattle, WA, United States of America, **2** Department of Global Health, University of Washington, Seattle, WA, United States of America, **3** Department of Epidemiology, University of Washington, Seattle, WA, United States of America, **4** Division of Allergy and Infectious Diseases, Department of Medicine, University of Washington, Seattle, WA, United States of America, **5** Department of Laboratory Medicine and Pathology, University of Washington, Seattle, WA, United States of America, **6** Vaccine and Infectious Disease Division, Fred Hutchinson Cancer Research Center, Seattle, WA, United States of America, **7** Benaroya Research Institute, Seattle, WA, United States of America

* amesic@uw.edu

**Data Availability Statement:** We are only presenting on data from peer-reviewed literature, not conducting out own primary data analysis. All

## Abstract

The interferon (IFN) system is a potent line of defense against viral infections. IFN-based agents already tested may be of use in COVID-19 or future viral respiratory outbreaks. Here we review the comparative efficacy, safety/tolerability, and future potential of IFN-based therapeutics. We reviewed human studies in which IFN or IFN pathway-interacting agents were used for viral respiratory infections. We identified 977 articles, of which 194 were included for full-text review. Of these, we deemed 35 articles to be relevant. The use of IFN-based agents for pre-exposure prophylaxis (n = 19) and treatment (n = 15) were most common, with intranasal (n = 22) as the most common route. We found IFN-α (n = 23) was used most often, and rhinovirus (n = 14) was the most common causative agent. Studies demonstrated mixed efficacy but generally positive safety and tolerability. Host-directed therapies, such as IFN or IFN inducers, are worthy of additional research to target viral respiratory infections lacking direct-acting antivirals.

## Introduction

The COVID-19 pandemic has significantly impacted nearly all aspects of people's lives globally, and highly effective agents for treatment of COVID-19 are still limited. Early in an epidemic, while potent direct-acting antivirals or antibody therapies are under development, generalizable agents could play a crucial role in epidemic control. One of the major lines of defense humans possess against viral infections is the interferon (IFN) system. IFNs are part of the innate immune system, and can control viral infection even in the absence of acquired immunity [1]. With most human cells having IFN receptors, IFNs are capable of inducing

data is available through an academic library service.

**Funding:** This work was funded by the Bill & Melinda Gates Foundation (grant number OPP1155935). This provided funding for four of the authors (Aldina Mesic, Emahlea Jackson, Mathias Lalika, and Rena Patel). David Koelle did not receive any funding for this work. The U.S. National Institutes of Health supported Ms. Aldina Mesic's and Dr. Rena C. Patel's efforts (D43-TW007267 and R25-TW009345 from the Fogarty International Center and K23AI120855, respectively). The funders had no role in study design, data collection and analysis, decision to publish, or preparation of the manuscript. https://www.gatesfoundation.org/ https://www.nih.gov/

**Competing interests:** The authors have declared that no competing interests exist.

expression of several hundred IFN-stimulated genes, encoding proteins that block viral replication, boost acquired immunity, or trigger apoptosis of infected cells [2].

IFNs are grouped into three distinct types, type I IFNs (IFN-α, IFN-β, IFN-κ, IFN-ω, IFN-ν); type II IFNs (IFN-γ); and type III IFNs (IFN-λ), each with features that make them a powerful defense system against viral infections [3]. Type II IFNs are only secreted by a specific subset of leukocytes, whereas type I and type III IFNs can be produced by both immune and parenchymal cells [4, 5]. Type III IFNs are produced by epithelial cells even in the absence of a strong inflammatory response [4, 6]. Similarly, toll-like receptor (TLR) agonists can also be potentially useful as a first line of defense against an emerging pandemic [7–9]. Before IFNs are expressed, the innate immune system detects the presence of chemical substances in pathogen structures or released by pathogens through an array of protein pattern-recognition receptors [2]. TLRs are exemplar pattern-recognition, and currently, 10 human TLRs have been well characterized [10].

Additionally, IFNs are related to viral respiratory disease susceptibility and progression. Recent studies have shown that IFNs induce a cytokine storm consisting of interleukin (IL)-1β, IL-6 and TNF-α—often markers of progression from mild or moderate to severe COVID-19 infection [11–13]. Individuals with IFN deficiency or autoantibodies that neutralize IFN appear to be more susceptible to severe COVID-19 [14]. This indicates that progression of viral respiratory infections from mild to severe forms may be associated with impaired host IFN response. Similarly, many viruses express proteins that suppress IFN induction or evade IFN-mediated responses [11, 15]. For instance, SARS-CoV-2, the causative agent of the ongoing COVID-19 pandemic, has developed the ability to suppress IFN induction, particularly type I IFNs [12].

SARS-CoV-2 is susceptible to the antiviral activities of type I and type III IFN, both in vitro and in vivo [16]. A recent study suggests that inhaled, nebulized IFN-β-1a may be safe and efficacious in the treatment of early and mild COVID-19 [17]. Furthermore, there are at least ten clinical trials underway that are testing the efficacy of IFN-based agents to prevent SARS-CoV-2 infection or disease progression [18].

If potent IFN-based agents, whether administered for prevention or early treatment of a viral infection, can be developed, and stored in sufficient quantities, the negative impact of future unrecognized viral respiratory epidemics may be mitigated early in the response while pathogen-specific agents are being developed. Given the potential role IFN-based agents may play in the mitigation of future viral respiratory outbreaks, our overall goal was to conduct a scoping review to summarize the literature on the use of IFN, IFN-stimulating agents, and TLR-agonists as pre-exposure prophylaxis, post-exposure prophylaxis and treatment of viral respiratory infections. To the best of our knowledge, no such scoping or systematic review has been conducted. Prior reviews have evaluated interferon for other disease areas (e.g., Hepatitis C, multiple sclerosis) [19, 20]. Several have evaluated the impact of certain types of interferon on COVID-19 such as IFN-β [21, 22] and Type 1 interferons [23]. Several reviews of IFN-α and IFN-β for the treatment of COVID-19 are ongoing [24]. A recent systematic review by Saleki *et al.* evaluated interferon for SARS, MERS, and COVID-19 [25]. Our scoping review is markedly different given the focus on any viral respiratory infections to develop knowledge for future pandemics with an unknown pathogen. We conducted this review to systematically map the existing literature on comparative efficacy and safety/tolerability of IFN-based agents in humans in which IFN or TLR-agonist-based agents may play a critical role in enhancing the host antiviral response, reducing viral shedding, and preventing severe disease—and thus the need for hospitalization—in patients with an early viral respiratory infection.

## Materials and methods

This scoping review aims to identify and synthesize existing published literature on comparative efficacy and tolerability of topical (intranasal or inhaled) versus injectable IFNs, type I versus type III IFNs, and IFN versus IFN-stimulating agents in humans. This scoping review was registered on Open Science Framework (DOI: 10.17605/OSF.IO/9G3SH) and was conducted between October 2020 and February 2021. We conducted this review in accordance with the PRISMA guidelines for scoping reviews [26] (see **S1 Table**) and the Arksey and O'Malley framework, a widely used framework for scoping studies [27]. A scoping review is appropriate here as we sought to characterize the body of evidence on IFN-agents for viral respiratory infection through a set of broad research questions [28]. A scoping review's approach is markedly different from that of a systematic review, which centers around narrow research questions, and involves the reviewers attempting to not only characterize but also assess the quality of the existing evidence. Thus, in this scoping review, we did not focus on qualifying the rigor or validity of the included studies and did not conduct a critical appraisal of the individual sources of evidence (an optional step in the PRISMA guidelines for scoping review) [26].

### Research question identification

The specific research questions that guided this work were:

- What is known about the mode (topical vs. systemic) and timing (pre- vs. post-exposure prophylaxis vs. treatment) of administration of IFN-based agents, and if any differences emerge for direct IFN vs. TLR-based agents?

- What is known about the efficacy of these agents, including the location of response (local vs. systemic) and pharmacodynamic biomarkers offering insights into how these agents may succeed?

- What differences exist between Type I and Type III IFNs in response to viral infections in the respiratory tract, and what IFN signatures may exist to predict severe disease or response to IFN-targeted therapies?

### Literature search

We conducted a complete systematic search of all relevant index peer-reviewed publications from inception to October 2020, and then repeated the search every four weeks to capture new studies from November to February 2021. Academic databases included: 1) PubMed/MEDLINE (National Library of Medicine); 2) EMBASE (Excerpta Medica hydroxyc); 3) Cochrane Library; and 4) medRxiv. medRxiv, a website for distributing unpublished manuscripts in medicine, health sciences, and clinical research, was utilized to capture studies on interferon and COVID-19 that were in the process of peer-review and publication. We included medRxiv as the literature on interferon and COVID-19 evolved so rapidly during the study period (October 2020-February 2021) and we did not want to miss important studies due to long peer review and publication processes during the ongoing pandemic.

Search terms focused on human clinical trials with any IFN or TLR across several viral infections and disease areas. We have included a summary of the search terms, the full search string, and detailed Medical Subject Headings (MeSH) in the **S2 Table**. A health sciences librarian systematically searched all four databases, and a team member imported all articles into Zotero (Roy Rosenzweig Center for History and New Media, version 5.0.94) for deduplication. We then imported all articles into REDCap electronic data capture tools hosted at University of Washington for abstract screening [29].

## Study selection

Three authors (AM, EJ, and ML) screened all titles and abstracts for inclusion/exclusion criteria (S3 Table), aiming to ascertain relevance to the research questions. We included studies that were published in English, peer-reviewed, have full text accessible using a library service, reported primary data, and conducted among humans (i.e., we excluded animal studies). We did not exclude studies with children. We excluded commentaries, clinical trial protocols, and any articles that did not report primary data, such as modeling studies or systematic reviews. Lastly, one author (AM) conducted a quality check of a random 5% of the articles to confirm inclusion/exclusion; this quality check yielded two discrepancies, which we resolved by discussion. Three authors (AM, EJ, and ML) were involved with full-text screening for inclusion/exclusion. A primary review was conducted by one author (EJ and ML) and a secondary review was conducted by the lead author (AM). We report study selection results per PRISMA guidelines in Fig 1.

## Data extraction

Once we selected articles for inclusion, we then randomly assigned articles for data extraction for two authors (EJ and ML) to conduct. A secondary data extraction was conducted by the lead author (AM). Generally, we extracted data on article information (i.e., authors, year, journal), methods, and outcomes. Specifically, we extracted detailed information related to agent administration including which agent and type administered, mode of administration, frequency, and duration. We assessed the timing of administration as: pre-exposure prophylaxis (PrEP), disease prevention before an individual is exposed to a disease-causing agent; post-exposure prophylaxis (PEP), disease prevention after an individual has a possible exposure to the disease-causing agent but before disease onset; and lastly, as treatment to individuals with a disease. We assessed efficacy as it was reported by the authors, which included clinical, virological, and/or patient-reported outcomes. A final review of the results (e.g., the extracted data) was conducted by four authors (AM, EJ, ML, and RP) together. Detailed information on all variables extracted can be found in the S4 Table.

## Results

We identified 1,248 articles meeting our inclusion criteria based on title and abstract content. After deduplication, there were 977 articles remaining. We excluded articles that were not written in English or irrelevant (i.e., not related to administration of IFN) (n = 24 and n = 720, respectively). After abstract review, we included 233 articles for full-text review, of which 198 additional articles we excluded for various reasons: 32 were ongoing clinical trials, four were protocols, 26 were review articles, 23 were missing full text, 22 were duplicates, five were non-English, 84 were not relevant (e.g., not focused on viral respiratory infection), and two were withdrawn articles.

During the full-text review, we included 35 peer-reviewed articles, most of which were published between 1981 and 1990 (n = 16), followed by 2020–2021 (n = 11). Although we had initially aimed to only include clinical studies, we included two observational studies from medRxiv, as they were highly relevant during the ongoing pandemic [30, 31]. Nearly all studies (n = 34) recruited adults aged 18 and above, whereas one study recruited infants aged 0–12 months [32]. Studies took place across 10 countries, and one study took place in multiple countries [33]. Most studies were randomized (n = 28), placebo-controlled (n = 25), and double-blinded (n = 26). Challenge studies comprised 11 of the total 35 studies.

PrEP and treatment were the most common timing indication of administration (n = 19 and n = 15, respectively). Among studies which used IFN-based agents for treatment, about

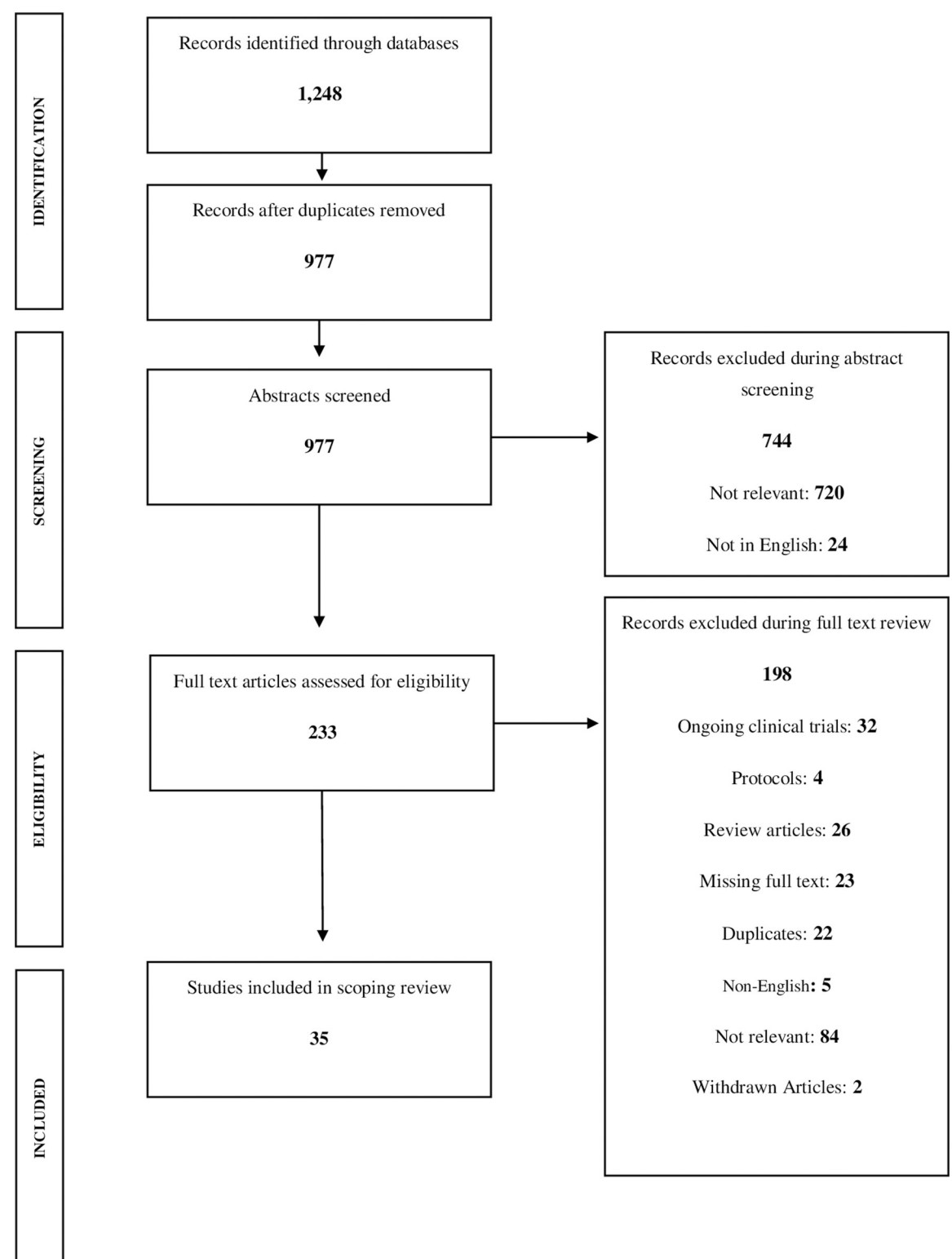

**Fig 1. PRISMA study selection procedure flow chart.**

half were among patients hospitalized with a viral respiratory illness (n = 8) and the other half were among patients not requiring hospitalization. Most studies administered IFN-based agents intranasally (n = 22), followed by subcutaneously (n = 5) and inhalation (n = 6). The most common pathogen was rhinovirus (n = 14), but many studies assessed both rhinovirus and other respiratory pathogens, including respiratory syncytial virus (n = 6) and influenza viruses (n = 8). Studies administered IFN-α most commonly (n = 23). We have included additional details related to the articles in **Table 1.**

## Pre-exposure prophylaxis results

**IFN-α.** IFN-α was often given as PrEP for rhinovirus infection and by the intranasal route (n = 9 and n = 14, respectively). The frequency of administration varied from once daily to four times daily for a minimum of four days and a maximum of 16 weeks.

IFN-α was shown to be safe and tolerated by participants. The most reported side effects were nasal discomfort and obstruction and blood-tinged mucus. Some studies reported significantly higher rates of side effects in the IFN-α treatment groups, particularly with higher doses, as compared to the control group [34–37].

Several studies detected efficacy of IFN-α, including a reduction in the incidence of respiratory infections [34, 35, 38–42]. Two additional studies confirmed a dose-response with only the highest dose of IFN protecting against common rhinoviruses and coronavirus infections compared to lower dose or control group [42, 43]. Several reported a reduction of symptom count [34, 40, 42, 44], severity [37, 38], and duration of illness [37]. However, in other studies, IFN-α was neither effective at preventing infections caused by respiratory syncytial virus (RSV), parainfluenza, or influenza viruses [35, 39, 45] nor did it reduce severity or duration of symptoms [37, 46]. One emerging theme from these studies is that IFN-α may have differential efficacy against different respiratory pathogens such that may be challenging to extrapolate from past studies to emerging viral pathogens.

**IFN- β.** Two studies tested IFN-β as PrEP for human rhinovirus infection. IFN-β was administered nasally near daily for about a month and was safe and tolerated [47, 48]. There were mixed findings regarding side effects. One study found no differences in nasal symptoms between the placebo and intervention groups after a rhinovirus challenge, but differences were found in subepithelial lymphocytes in nasal biopsy specimens and in self-reported mucosal bleeding in the high dose group compared to the low-dose and placebo groups after viral challenge [47]. The second study did not find any differences between treatment and control groups [48]. Neither study found IFN-β to be effective in reducing the occurrence of symptoms, illness frequency, or number of days with subjective cold-like symptoms after a viral challenge.

**TLR-agonist.** One study administered a TLR-3 agonist, polyinosinic-polycytidylic acid, intranasally and daily for six days [49]. The study noted no toxic effects, indicating the treatment was safe. The article included three viral challenge studies using human rhinovirus 13 or type A2 influenza virus/Hong Kong/68 virus and found evidence for a reduction in symptoms of upper respiratory tract illness. One study showed a reduction in viral shedding [49].

**IFN-stimulating agent.** An IFN-stimulating agent, N,N-dioctadecyl-N'-bis-hydroxyethyl)-propanediamine, was administered as PrEP and given three times on the day before a rhinovirus challenge and then four and eight hours after the challenge [50]. Although the study did not report safety or tolerability outcomes, it reported no difference in rates of infection or incidence of illness with the treatment. However, there were indications of faster clinical recovery when higher titers of IFN were found in nasal secretions. These data suggest, but do not prove, that IFN or IFN-induced genes may have mediated a shorter symptom duration. We have included additional details in **Table 2.**

**Table 1. Studies included in scoping review (N = 35).**

| Variable | N (%) |
|---|---|
| **Year of publication** | |
| 1970–1980 | 2 (6%) |
| 1981–1990 | 16 (46%) |
| 1991–2000 | 1 (3%) |
| 2001–2010 | 1 (3%) |
| 2011–2019 | 2 (6%) |
| 2020–2021 | 11 (31%) |
| Pre-print | 2 (6%) |
| **Population[B]** | |
| Healthy adults | 21 (54%) |
| Adults with viral respiratory illness not requiring hospitalization | 7 (19%) |
| Adults with viral respiratory illness requiring hospitalization | 8 (22%) |
| Families (age not specified) | 1 (3%) |
| Infants (0–12 months) | 1 (3%) |
| **Country** | |
| Australia | 1 (3%) |
| Canada | 1 (3%) |
| China | 8 (22%) |
| Cuba | 1 (3%) |
| Hong Kong | 1 (3%) |
| Iran | 1 (3%) |
| Saudi Arabia | 1 (3%) |
| Switzerland | 1 (3%) |
| United Kingdom | 6 (17%) |
| United States | 13 26%) |
| Global | 1 (3%) |
| **Randomization** | |
| Randomized | 28 (80%) |
| Not randomized | 5 (4%) |
| Not specified | 4 (11%) |
| **Placebo** | |
| Placebo-controlled | 25 (71%) |
| No placebo | 10 (26%) |
| **Blinding** | |
| Double-blinded | 26 (74%) |
| Open-label/no blinding | 8 (23%) |
| Not specified | 1 (3%) |
| **Challenge Study** | |
| Challenge study (e.g., participants challenged with a virus) | 11 (31%) |
| Not a challenge study | 24 (69%) |
| **Type of Study** | |
| Clinical Trial | 33 (97%) |
| Observational | 2 (5%) |
| **Timing of administration** | |
| Pre-exposure prophylaxis | 19 (54%) |
| Post-exposure prophylaxis[A] | 2 (6%) |
| Treatment[A] | 15 (43%) |

(*Continued*)

**Table 1.** (Continued)

| Variable | N (%) |
|---|---|
| **Mode of administration[B]** | |
| Intranasal | 22 (63%) |
| Inhaled | 6 (17%) |
| Intramuscular | 2 (6%) |
| Intravenous | 1 (3%) |
| Subcutaneous | 5 (14%) |
| Oral | 1 (3%) |
| **Pathogen or clinical condition indication[B]** | |
| Adenovirus | 5 (14%) |
| Bronchiolitis | 1 (3%) |
| Influenza | 8 (23%) |
| Middle east respiratory syndrome | 1 (3%) |
| Respiratory syncytial virus | 6 (17%) |
| Rhinovirus | 14 (40%) |
| Seasonal coronavirus colds | 2 (6%) |
| Severe acute respiratory syndrome coronavirus 2 | 10 (29%) |
| Upper respiratory tract illness | 2 (6%) |
| **Agent type** | |
| IFN-α[A] | 23 (66%) |
| IFN-β | 9 (26%) |
| IFN-κ | 1 (3%) |
| IFN-λ | 1 (3%) |
| IFN-stimulating agents | 1 (3%) |
| Toll-like receptor-agonists | 1 (3%) |

A. One study gave IFN-α as treatment to the index case of cold, and as PEP to the household members without illness.

B. Multiple categories possible for each article.

Acronyms: IFN: interferon.

## Post-exposure prophylaxis results

**IFN-α.**    Two studies administered IFN as PEP. Herzog *et al*. administered IFN-α for the prevention of cold symptoms caused by rhinovirus, adenovirus, influenza virus A/B, parainfluenza virus ½/3, or RSV [53]. IFN-α was given to household members within two days after the appearance of a common cold in the household. Although the treatment was found to be well tolerated, it did not prevent cold symptoms at either dose (1.5 x 10$^6$ IU or 0.3 x 10$^6$ IU). However, it did significantly reduce symptom duration and severity.

**IFN- β.**    Sperber *et al*. administered IFN-β three times daily for 4.3 days. Intranasal administration was considered safe and well tolerated with no differences between the control and treatment groups. Illness rates did not vary, although the frequency of virus shedding was reduced on the fourth and sixth post day post-challenge in the IFN-β group [54]. We have included additional details in **Table 3**.

## Treatment results

**IFN-α.**    Seven studies administered IFN-α as a treatment for a viral respiratory infection, and three of these focused on treating SARS-CoV-2. Other studies included diseases, such as

**Table 2. Pre-Exposure prophylaxis studies (N = 19).**

| Study | Study Type | Population; Country | Indication | Agent; Mode | Frequency; Duration; Dose[A] | Safety & Efficacy |
|-------|-----------|--------------------|-----------|------------|-----------------------------|-------------------|
| | | | | IFN-α | | |
| Bennett, 2013 [51] | Randomized; Double-blind; Placebo-controlled | 200 healthy adults; Australia | Influenza B, adenovirus, respiratory syncytial, virus and parainfluenza | 3:1 ratio of IFN-α-2b and IFN-α-8; Oral | Once daily; 16 weeks | *Safety*: Similar adverse event rate between treatment and control groups. Most common symptoms included hay fever, diarrhea, and headache. About 89% of participants completed 16 weeks of treatment. *Efficacy*: Did not reduce incidence of impact of acute respiratory illness or the impact on daily activities. Post-hoc analyses indicate efficacy in some groups (males, those >50 years, those that had received the seasonal flu vaccine). |
| Farr, 1984 [34] | Randomized; Double-blind; Placebo-controlled | 304 healthy adults; United States | Rhinovirus | IFN-α-2; Intranasal | Once daily; 22 days | *Safety*: Nasal symptoms (obstruction, discomfort, blood-tinged mucus) and signs (punctuate bleeding, erosions, ulcerations) occurred significantly more often in treatment than control participants. *Efficacy*: Statistically significant reduction in the incidence of respiratory infection (measured by rhinovirus isolation) and symptoms of tracheobronchitis compared to placebo group. |
| Gao, 2010 [35] | Randomized; Double-blind; Placebo-controlled | 1,449 male military recruits (age 16–23); China | Influenza A virus, influenza B virus parainfluenza viruses 1–3 and adenovirus species B | IFN-α-2b; Intranasal | Twice daily; 5 days | *Safety*: No participants withdrew from the study. Initial statistically significant differences in side effects (e.g., dry pharynx, epistaxis) between the treatment and control groups declined after the first five days of treatment. *Efficacy*: Statistically significant prevention of infections caused by influenza A virus, influenza B virus, parainfluenza viruses 1–3, and adenovirus species B (measured by viral isolation) but not respiratory syncytial virus. |
| Hayden, 1983 [52] | Randomized; Double-blind; Placebo controlled; Challenge | Two studies with 26 and 29 volunteers; United States | Rhinovirus | IFN-α-2a; Intranasal | Four times daily; 4 days | *Safety*: No abnormalities in hematological, liver, or renal function tests besides three participants developing transient leukopenia. *Efficacy*: Two challenge studies found efficacy rates of multi-dose IFN for prevention of infection, virus shedding, and respiratory syncytial virus-specific colds were 78%, 78%, and 100%, respectively. One dose rates were 45%, 64%, and 75%, respectively. |

*(Continued)*

**Table 2.** (*Continued*)

| Study | Study Type | Population; Country | Indication | Agent; Mode | Frequency; Duration; Dose[A] | Safety & Efficacy |
|-------|-----------|---------------------|------------|-------------|------------------------------|-------------------|
| Higgins, 1990 [38] | Randomized; Double-blind; Placebo-controlled; Challenge | Two studies, one with 46 healthy adults and another with 60 healthy adults; United Kingdom | respiratory syncytial virus | IFN-α-2a; Intranasal | Three times daily; 4·3 days | *Safety*: Two withdrawals, no safety outcomes reported. *Efficacy*: Incidence and severity of signs and symptoms was reduced in treatment group. In a second study with those that developed colds, IFN-α-2a did not reduce symptom severity or duration. |
| Meng, 2021 [44] | Not randomized; Open-label; No placebo; Prospective | 2,944 healthy medical staff selected based on level of exposure to COVID-19; China | SARS-CoV-2 | IFN-α-2a; Intranasal | Four times daily; 28 days | *Safety*: Transient irritation of the nasal mucosa. *Efficacy*: No reports of COVID-19 pneumonia or, fever/respiratory symptoms among low- and high-risk healthcare workers in Hubei province, China. Results were compared to a control group of other healthcare workers in Wuhan which had 2,035 reported cases, indicating efficacy. |
| Monto, 1986 [39] | Randomized; Double-blind; Placebo-controlled | 400 healthy college students; United States | Rhinovirus | IFN-α-2b; Intranasal | Twice daily; 4 weeks | *Safety*: Treatment well-tolerated although side effects (i.e., blood-tinged mucus, dry nose, ulcers/erosions) were often reported. *Efficacy*: Rhinovirus infections were prevented (76% protective efficacy) but parainfluenza infections were not. Parainfluenza symptoms were significantly reduced. |
| Monto, 1988 [45] | Randomization not specified; Double-blind; Placebo-controlled | 600 and 350 healthy college students; United States | Influenza, rhinovirus | IFN-α-2b; Intranasal | Twice daily or once daily; 4 weeks; $1\cdot7 \times 10^6$ IU; $2\cdot5 \times 10^6$ IU; $3\cdot0 \times 10^6$ | *Safety*: Common side effects reported included blood-tinged mucus and dry nose. Side effects were dose and frequency related. *Efficacy*: Efficacy was dose-dependent. Daily doses of $3\cdot0 \times 10^6$ IU had a 76% efficacy in preventing rhinovirus infections. $2\cdot5 \times 10^6$ IU was still statistically significant while $1\cdot7 \times 10^6$ IU had a minimal impact on rhinovirus infection (efficacy: 22–27%). |
| Phillpotts, 1984 [40] | Not randomized, matched by sex/age; Double-blind; Placebo-controlled, Challenge | Two studies, one with 44 healthy adults and another with 30 healthy adults; United Kingdom | Influenza, Rhinovirus | IFN-α-n1; Intranasal | Three times daily; 4·3 days | *Safety*: No evidence of intolerance and no differences in hematological and biochemical tests. A few participants developed mild nasal symptoms. *Efficacy*: Challenge study found statistically significant lower incidence of human rhinoviruses 9 and 14 and influenza virus A/Eng/40/83 disease (measured by clinical scores). Mean daily and total clinical scores and mean daily and total nasal secretion weights were significantly greater in the control than the treatment group. These parameters were generally improved after influenza challenge but only on day 2. |

(*Continued*)

**Table 2.** (Continued)

| Study | Study Type | Population; Country | Indication | Agent; Mode | Frequency; Duration; Dose[A] | Safety & Efficacy |
|---|---|---|---|---|---|---|
| Samo, 1984 [41] | Randomized; Double-blind; Placebo-controlled; Challenge | Two studies (one challenge study and one tolerability) with 63 and 59 healthy adults respectively; United States | Rhinovirus | IFN-α-A; Intranasal | Two doses daily; 4 days, 26 days for the tolerance study; $0.7 \times 10^6$ IU $2.4 \times 10^6$ IU | *Safety*: In the four-day study, no local or systemic reactions were reported. In 26-day tolerance study, 15% of participants reported local adverse reactions (e.g., bloody mucus, nasal mucosal erosions). *Efficacy*: Challenge study found a dose-response effect. After four days of administration of $2.4 \times 10^6$ IU led to significantly lowered frequency of illness versus control but $0.7 \times 10^6$ IU was not effective. |
| Sarno, 1983 [42] | Randomized; Double-blind; Placebo-controlled; Challenge | 26 healthy adults; United States | Rhinovirus | IFN-α-A; Intranasal; | Daily; 4 days; $10 \times 10^6$ units | *Safety*: About 26% (n = 5) if the participants treated with IFN reported blood in the nasal mucus and superficial nasal erosions. *Efficacy*: There were significant reductions in illness frequency, mean symptom score, nasal secretion weights, and frequency of virus isolation in the IFN group compared to the control group. |
| Scott,1983 [43] | Randomized; Double-blind; Placebo-controlled | 68 healthy adult volunteers; United Kingdom | Common cold (defined as upper respiratory symptoms) | IFN; Intranasal | Twice daily; 28 days; 0.44 Mu, 1.2 Mu, 4.4 Mu | *Safety*: No differences in discontinuation due to upper respiratory symptoms between the IFN groups and control group. The highest dose caused upper respiratory symptoms (e.g., dry nose, bleeding) *Efficacy*: The lowest dose did not protect against clinical colds whereas the highest dose seemed to have protected against common rhinovirus and coronavirus colds (i.e., there were no reported colds or viruses isolated) |
| Higgins, 1988 [46] | Randomized; Double-blind,' No placebo; Challenge | Two studies (tolerance and challenge) with 11 and 51 healthy adult volunteers; United Kingdom | Rhinovirus 9 and 14 | IFN- α-2a; Intranasal | Four times daily; 4.25 days | *Safety*: One participant withdrew from the study. Side effects reported included nasal singing and eye watering. *Efficacy*: Challenge study found that IFN did not reduce the severity of colds (RV9 and RV14) and was not enhanced by the administration of enviroxime. |
| Tannock, 1988 [36] | Randomized; Double-blind; Placebo-controlled | 412 healthy adult volunteers; Australia | Influenza, Human rhinovirus, respiratory syncytial virus, adenovirus | IFN- α-2a; Intranasal | Once daily; 28 days | *Safety*: Blood-tinged mucus and nasal stuffiness occurred more frequently with higher doses of IFN. *Efficacy*: There was no statistically significant difference in incidence of laboratory proven viral infections. |

(*Continued*)

**Table 2.** (Continued)

| Study | Study Type | Population; Country | Indication | Agent; Mode | Frequency; Duration; Dose[A] | Safety & Efficacy |
|---|---|---|---|---|---|---|
| Turner, 1986 [37] | Randomized; Double-blind; Placebo-controlled; Challenge | 56 healthy adult college students; United States | Coronavirus | IFN-α-2b; Intranasal | Twice daily; 15 days | *Safety*: No significant differences in bloody nasal mucus and nasal or total symptom scores. Nasal bleeding sites were found significantly more often in the treatment group. There were no abnormalities in labs. *Efficacy*: Challenge study with coronavirus 229E strain found that IFN significantly shortened the duration and severity of cold symptoms. IFN did not reduce the incidence of infection. |
| | | | | IFN-β | | |
| Sperber, 1988 [47] | Randomized; Double-blind; Placebo-controlled; Challenge | 120 healthy adults; United States | Human rhinovirus | IFN-β-serine; Intranasal | Once daily; 25 days; $3 \times 10^6$ U, $12 \times 10^6$ U | *Safety*: There were no differences in nasal symptoms. There were differences in subepithelial lymphocytes and mucosal bleeding sides in the high dose group versus the low dose and placebo group. *Efficacy*: In challenge study with rhinoviruses, there was no statistically significant differences in the occurrence of symptoms. |
| Sperber, 1989 [48] | Randomized; Double-blind; Placebo-controlled | Two studies with 296 and 383 healthy adults; United States | Human rhinovirus | IFN-β-serine; Intranasal | Once daily (except Sundays); 4 weeks; $9 \times 10^6$ units, $24 \times 10^6$ units | *Safety*: Tolerance did not differ at either dosage or between treatment or control. Efficacy: Illness frequency and number of days with subjective colds did not differ between the treatment and control group. |
| | | | | TLR-Agonists | | |
| Hill, 1972 [49] | Randomization not specified; Placebo-controlled; Blinding not specified; Challenge | Two studies with 18 healthy volunteers each; United States | Human rhinovirus 13, type A2 influenza virus/Hong Kong/68 | polyinosinio-polycytidylic acid, 0.7 mg (0.01 mg/kg) for one day followed by 0.35 mg (0.005 mg/kg) or 7 mg (0.1 mg/kg) for one day followed by 3.5 mg (0.05 mg/kg); Intranasal | Daily; 6 days | *Safety*: No toxic effects were detected. *Efficacy*: In three challenge studies, there was a reduction in symptoms of upper-respiratory tract illness. Poly I.poly C reduced virus shedding in one of the two rhinovirus studies. The treatment did not reduce virus shedding of rhinovirus and type A2 influenza virus in the other two trials. |
| | | | | IFN-stimulating agents | | |
| Gatmaitan, 1973 [50] | Randomization not specified; Placebo-controlled; Double-blind; Challenge | 38 healthy college volunteers; United States | Human rhinovirus | N,N-dioctadecyl-N'-bis-(hydroxyethyl)-propanediamine; Intranasal | Three times daily on day the before the challenge, two more doses given 4 and 8 hours after the challenge; 2 days | *Safety*: No safety outcomes reported. *Efficacy*: There was no difference in rate of infection or incidence of illness. There was a faster recovery from illness with the appearance of high titers of interferon in nasal secretions. |

Dose is only reported for studies which administered more than one dose to participants.

Acronyms: IFN: interferon.

**Table 3. Post-Exposure prophylaxis studies (N = 2).**

| Study | Study Type | Population; Country | Indication | Agent; Mode | Frequency; Duration; Dose[A] | Safety & Efficacy |
|---|---|---|---|---|---|---|
| | | | IFN-α | | | |
| Herzog, 1986 [53] | Randomized; Double-blind; Placebo-controlled | 191 families (587 participants, age not specified, one index case of illness in each family); Switzerland | Rhinovirus, Adenovirus, Influenza Virus A/B, Parainfuenza Virus ½/3, Respiratory Syncytial Virus | IFN-α-A; Intranasal | Twice daily; 5 days, starting within 2 days after appearance of self-reported common cold in household; 1·5 x 10⁶ IU, 0·3 x 10⁶ IU | *Safety*: Treatment was well-tolerated. *Efficacy*: Both doses did not prevent common colds. 1·5 x 10⁶ IU did statistically significantly shorten the duration and reduce the severity of colds. |
| | | | IFN-β | | | |
| Sperber, 1992 [54] | Randomized; Double-blind; Placebo-controlled; Challenge | 40 healthy adults; United States | Rhinovirus | IFN-β-serine; Intranasal | Three times daily; 4·3 days, beginning 36 hours after infection; 12 x 10⁶ U | *Safety*: Nasal bleeding, burning, and dryness were reported equally between treatment and control groups. No lab abnormalities were found. *Efficacy*: In this rhinovirus challenge study, illness rates and severity did not differ between the treatment and control groups. Virus shedding was reduced on the fourth and sixth post challenge day among IFN recipients. Middle ear dysfunction associated with colds (i.e., abnormal eustachian tube function, middle-ear pressure abnormalities) were reduced in the treatment group. |

Acronyms: IFN: interferon.

non-influenza viral pneumonia, and upper respiratory tract illness, common cold, and infant bronchiolitis, that were not virologically defined. The mode of administration varied across studies, with two administering treatment intranasally [53, 55], three by inhalation [32, 56, 57], and one intramuscularly [30]. One study did not report the route of administration [31]. Treatment frequency and duration ranged from four times daily to three times weekly and 5–17 days, respectively.

Most studies reported mild side effects. Only two of seven studies reported differences in side effects between the treatment and control groups, including for self-reported blood in the nasal mucus (56) and higher counts of adverse events [56].

Regarding efficacy, the proportion of patients discharged was higher and case fatality rate lower in the IFN-α vs. control group in SARS-CoV-2 [30]. Early administration (less than or equal to five days after admission) was associated with reduced in-hospital mortality whereas late administration increased mortality. However, another study failed to find any statistically significant differences between groups for viral shedding or illness progression in SARS-CoV-2 [57].

For non-influenza viral pneumonia, IFN-α-1b led to an improvement in the overall response rate for expectoration and pulmonary rates, but not for chest pain, respiratory rate, or coughing [56]. IFN-α did not affect the severity or duration of upper respiratory syndromes of unknown etiology (i.e. common colds) [53] or affect respiratory symptoms scores or resolution of symptoms [55]. IFN-α improved coughing severity and reduced wheezing among infants aged 0–12 months with bronchiolitis (which is often caused by viruses, such as RSV) [32]. One study reported early admission (5 or less day after admission) was associated with

reduced in-hospital mortality whereas late administration (more than five days after admission was associated with increased mortality and delayed recovery [31].

**IFN- β.**    Six studies administered IFN-β as a treatment by the subcutaneous route [33, 58–60], by the inhalation route [17, 61], and by the intravenous route [33]. Most studies administered IFN-β daily [17, 33, 61] or every alternate day [58–60].

Treatment with IFN-β was well tolerated across studies with no differences in adverse events between the control and treatment groups. For hospitalized patients with Middle East Respiratory Syndrome, treatment with IFN-β and lopinavir-ritonavir led to significantly lower mortality than placebo (of note, there was no IFN-β alone comparison group) [58]. For SARS-CoV-2, there were mixed efficacy findings. Hung *et al.* found that patients treated with a triple combination IFN-β, lopinavir–ritonavir, and ribavirin had significantly shorter median time from treatment start to negative nasopharyngeal swab detection compared to the control group [59]. Similarly, Monk *et al.* found greater odds of clinical improvement [17], and Rahmani *et al.* observed a shorter time to clinical improvement for the IFN-β when compared to the control groups of lopinavir/ritonavir or atazanavir/ritonavir plus hydroxychloroquine [60]. However, IFN-β did not affect the duration of hospitalization [33, 60], initiation of ventilation [33], or duration of intensive care unit stay [60].

**IFN-κ.**    One study administered IFN-κ in combination with trefoil factor 2 to treat SARS-CoV-2 and given three time every 48 hours for 10 days [62]. The combination treatment led to a shorter median improvement time, median time to cough relief, viral RNA reversion, and median hospital stay duration compared to the control group [62].

**IFN-λ.**    One dose of subcutaneous IFN-λ was given for SARS-CoV-2 and led to a faster decline of SARS-CoV-2 RNA shedding compared to the control group [63]. The IFN-λ group had a higher likelihood of viral clearance by day seven. We have included additional details in **Table 4**.

## Discussion

In this scoping review covering efficacy and safety/tolerability of IFN-based agents for viral respiratory infections, we report the following findings. Studies reported mixed efficacy, ranging from strong positive impacts to indifference to worsening of respiratory symptoms with treatment. The IFN-based agents were generally safe and well tolerated, particularly with inhaled or other topical administration of IFN. We found that studies of IFN-based agents for PrEP or treatment following symptom onset were more frequent than for PEP.

For IFN-based agents, efficacy seems to depend on the type of IFN-agent, frequency, duration, and timing of administration. There is a paucity of randomized control trials to help establish superiority amongst these factors. While the overall findings are mixed, the efficacy signals suggest a potential role that IFN-based agents may play for COVID-19 and other future pandemics. Studies to date suggest that IFN-β may be useful as treatment while IFN-α may play a larger role in PrEP for various viral respiratory infections, including COVID-19. Our findings are well aligned with other systematic reviews and meta-analysis results which suggest Type 1 interferons reduce mortality and increase faster hospital discharge among patients with COVID-19 [23], IFN-β reduces hospitalization length and respiratory symptoms of COVID-19 [21], and IFN-based agents lead to improvements in chest X-rays findings and lower levels of inflammatory cytokines when used in combination with antivirals, corticosteroids, and traditional medicine for SARS, MERS, and COVID-19 [25].

IFN-based agents were well-tolerated with favorable safety/tolerability profiles. Most studies administered agents intranasally, subcutaneously, or via inhalation. There were few study withdrawals reported due to side effects or serious or non-serious adverse events. Overall,

**Table 4. Treatment studies (N = 15).**

| Study | Study Type | Population; Country | Indication | Agent; Mode | Frequency; Duration; Dose[A] | Safety & Efficacy: |
|---|---|---|---|---|---|---|
| IFN-α | | | | | | |
| Chen, 2020 [32][D] | Randomized; Open-label; No placebo | 675 infants aged 0–12 months hospitalized with bronchiolitis; China | Infant bronchiolitis[A] | IFN-α-1b; Intramuscular; inhalation | Daily; 7 days; 1 microgram per kilogram or 2 microgram per kilogram; | *Safety*: No reports of related side effects, serious complications, recurrence of fever after abatement and no regression of mental status. Other side effects (poor mental status, rash, poor appetite) were rare and balanced across control and treatment groups. *Efficacy*: Coughing severity change was significantly different between the IFN nebulization 2 and control groups. Lowell wheezing score change between days 3–5 was different between the IFN nebulization 1 and control group. However, there were no differences regarding the number of consecutive days with fever, three-concave sign, sleepiness, and loss of appetite. |
| Hayden, 1988 [55] | Randomized; Double-blind; No placebo | 228 adults within 48 hours oof the onset of symptoms of upper respiratory tract illness; United States | Upper respiratory tract illness (defined as self-reported sneezing, rhinorrhea, nasal congestion, sore throat, hoarseness, cough) | IFN-α-1b, 10 or 20 microgram; Intranasal | Four times daily; 5 days | *Safety*: No differences in reports of nasal dryness or nasal burning between groups. Participants in the 10 MU and 20 MU IFN group were significantly more likely to report blood in the nasal mucus. *Efficacy*: Among patients with proven rhinovirus colds treated within 24 hours, the median duration was significantly longer in the 20 MU group compared to the control group. No differences were observed in respiratory symptom scores or resolution of specific symptoms. |
| Herzog, 1986 [53] | Randomized; double-blind,; Placebo-controlled | 191 families (587 participants, age not specified, one index case of illness in each family, which was treated with IFN-α); Switzerland | Rhinovirus, Adenovirus, Influenza Virus A/B, Parainfuenza Virus ½/3, Respiratory Syncytial Virus | IFN-α-A; Intranasal | Twice daily; 5 days | *Safety*: Treatment was well tolerated. No other safety outcomes were reported. *Efficacy*: IFN did not affect the severity or duration of flu-like illness. |
| Huang, 2020 [57] | Randomized; Open-label; No placebo | 101 adults hospitalized with mild/moderate covid; China | SARS-CoV-2 | IFN-α; Inhalation | Twice daily; 14 days | *Safety*: All groups received IFN in combination with other treatments (ribavirin, lopinavir/ritonavir). Commonly reported side effects included diarrhea, vomiting, and electrolyte disorders; adverse events did not differ by treatment group. *Efficacy*: There were no statistically significant differences between groups on nucleic acid negativity or illness progression. |
| Jiang, 2020 [56] | Randomized; Double-blind; Placebo-controlled | 164 adults hospitalized with noninfluenza viral pneumonia; China | Non-influenza viral pneumonia | IFN-α-1b; Inhalation | Once daily; 7 days | *Safety*: There were statistically significant differences in the number of adverse events reported (treatment group: 6·5%, control group: 3·5%, p<0.05). *Efficacy*: After seven days of treatment, the overall response rate (ORR) was improved in the IFN group. The ORRs for expectoration and pulmonary rates were higher in the IFN group than the control group. There were no differences in ORRs for chest pain, respiratory rate, or coughing. |

*(Continued)*

**Table 4.** (Continued)

| Study | Study Type | Population; Country | Indication | Agent; Mode | Frequency; Duration; Dose[A] | Safety & Efficacy: |
|-------|-----------|-------------------|-----------|-------------|------------------------------|--------------------|
| Pereda, 2020 [30] | Not randomized; No blinding; No placebo; observational | 814 adults with COVID-19; Cuba | SARS-CoV-2 | IFN-α-2b; Intramuscular | Three times weekly; Two weeks | *Safety*: IFN treatment was withdrawn from patients who progressed to critical disease. No other safety outcomes were reported. *Efficacy*: The proportion of patients discharged from the hospital was higher in the IFN group versus non-IFN. The case fatality was reduced from 2.95% for all patients to 0.92% for the IFN group. |
| Wang, 2020 [31] | Not randomized; No blinding; No placebo; observational | 446 adults hospitalized with COVID-19; China | SARS-CoV-2 | IFN-α-2b; Not reported | Not reported; 5–17 days | *Safety*: No safety outcomes were reported. *Efficacy*: Early administration (< = 5 days after admission) was associated with reduced in-hospital mortality whereas late administration was associated with increased mortality. Early administration was not associated with hospital discharge or computed tomography scan improvement but late administration was associated with delayed recovery. |
| **IFN-β** | | | | | | |
| Arabi, 2020 [58] | Randomized; Double-blind; Placebo-controlled | 95 adults hospitalized with MERS; Saudi Arabia | Middle East respiratory syndrome | IFN-β-1b; Subcutaneous | Every alternate day; 14 days; 0·25 mg [8 million IU] | *Safety*: The incidence of adverse events did not differ significantly between the treatment and control group. Serious adverse events were reported in 9% (n = 4) patients in the intervention group and 19% (n = 10) in the control group. *Efficacy*: Treatment with interferon and lopinavir-ritonavir led to significantly lower mortality than placebo among patients that were hospitalized. |
| Djukanovic, 2014 [61] | Randomized; Double-blind; Placebo-controlled | 147 adults with asthma; United Kingdom | Asthma and cold symptoms[B] | IFN-β-1a; Inhalation | Once daily; 14 days within 24 hours of developing a cold | *Safety*: Treatment was well-tolerated with no patients stopping treatment. Similar rates of treatment-emergent adverse events. *Efficacy*: Among participants with asthma and a cold caused by respiratory viruses, IFN did not have an effect on asthma symptoms. However, it did significantly enhance morning peak expiratory flow recovery, reducing the need for additional treatment. It also significantly enhanced innate immunity, measured by blood and sputum biomarkers. |
| Hung, 2020 [59] | Randomized; Open-label; No placebo | 127 adults with COVID; Hong Kong | SARS-CoV-2 | IFN-β-1b; group treated on day 7–14 of symptoms did not receive IFN-β-1b due to proinflammatory effects; Subcutaneous | Every alternate day; 14 days | *Safety*: Common adverse events included diarrhea and nausea with no significant differences between groups. *Efficacy*: Patients treated with a triple combination IFN, lopinavir–ritonavir, and ribavirin had significantly shorter median time from treatment start to negative nasopharyngeal swab compared to the control group. |
| Monk, 2020 [17] | Randomized; double-blind; Placebo-controlled | 101 adults hospitalized with COVID-19; United Kingdom | SARS-CoV-2 | IFN-β-1a; Inhalation | Once daily; 14 days | *Safety*: The treatment was well tolerated. The most common adverse event reported were headaches. Cough was the most common treatment-related adverse event. *Efficacy*: Patients receiving IFN had greater odds of improvement on the WHO Ordinal Scale for Clinical Improvement (OSCI) scale on day 15 and 16. Further, patients in the treatment group were more likely than those in the control group to recover to an OSCI score of 1 (indicating no limitation of activities). |

*(Continued)*

**Table 4.** (Continued)

| Study | Study Type | Population; Country | Indication | Agent; Mode | Frequency; Duration; Dose[A] | Safety & Efficacy: |
|---|---|---|---|---|---|---|
| Rahmani, 2020 [60] | Randomized; Double-blind; Placebo-controlled | 80 adults with severe covid; Iran | SARS-CoV-2 | IFN-β-1b; Subcutaneous | Every alternate day; 14 days | *Safety*: Adverse event incidence was higher in the control group. IFN-related adverse events included injection site reactions and flu-like syndrome. *Efficacy*: Time to clinical improvement was significantly shorter for the IFN than the control group (lopinavir/ritonavir or atazanavir/ritonavir plus hydroxychloroquin). Discharge by day 14 was significantly more likely in the IFN group. The ICU admission rate was significantly higher in the control group when compared to the IFN group. However, the duration of hospitalization and ICU stay were not significantly different. |
| Pan, 2021 [33] | Randomized; Open-label; No placebo | 11,330 adults hospitalized with COVID-19; global | SARS-CoV-2 | IFN-β-1a; Subcutaneous; intravenous | Subcutaneous: 3 doses; intravenous: daily; 6 days | *Safety*: No safety outcomes were reported. *Efficacy*: IFN alone, or in combination with lopinavir did not reduce mortality (overall or in any subgroup), initiation of ventilation, or hospital duration compared to when compared to remdesivir, lopinavir, and no treatment. |
| IFN-κ | | | | | | |
| Fu, 2020 [62] | Open-label, Non-randomized; No placebo | 33 adults hospitalized with COVID; China | SARS-CoV-2 | IFN-κ; Inhalation | Three times every 48 hours; 10 days | *Safety*: No treatment-associated severe adverse events were observed. No differences in safety evaluations between the treatment and control groups. *Efficacy*: Median improvement time was significantly shorter in the trefoil factor 2/IFN group compared to the control group. The treatment group had significantly shorter median time to cough relief, in viral RNA reversion, and in the median hospitalization stay duration. |
| IFN-λ | | | | | | |
| Feld, 2021 [63] | Randomized; Double-blind; Placebo-controlled | 60 adults with COVID; Canada | SARS-CoV-2 | IFN-λ; Subcutaneous | Once; N/A | *Safety*: Symptoms were mild in both groups with no difference in frequency, severity, or rate of improvement in any of the symptom categories between groups. *Efficacy*: The decline of SARS-CoV-2 RNA was greater in the IFN group compared to the control group. After controlling for baseline viral load, the IFN group treatment resulted in a significantly higher likelihood of viral clearance by day 7. |

A. This study was Ied as infant bronchiolitis is often caused by viruses (i.e., respiratory syncytial virus), meeting our eligibility criteria.

B. This study was included as cold symptoms are often caused by Coronaviruses, meeting our eligibility criteria.

C. Dose is only reported for studies which administered more than one dosage to participants.

Acronyms: IFN: interferon; ICU: intensive care unit.

intranasal dosing had the most favorable safety and tolerability profile and has the advantage of delivering drug locally.

Pharmacokinetic and pharmacodynamic outcome data, which inform frequency, duration, and timing of administration and illuminate the mechanisms of action for IFN-based agents to mitigate viral respiratory infections, were extremely limited. Only a few studies commented on

these measures and found that IFN-based agents appear to have a reasonable half-life (i.e., in terms of the durability of the administered IFN-α and the duration of downstream biological effects; data not presented due to the scarcity of findings) [34, 42, 47]. Thus, future trials in humans would benefit from inclusion of pharmacokinetic and pharmacodynamic outcomes to better inform existing knowledge gaps.

While our scoping review identified several human clinical trials with IFN-based agents for viral respiratory infections, there are still apparent gaps in the literature that require further study to determine the full role of IFN-based agents in mitigating future viral respiratory epidemics. We highlight these gaps as: 1) lack of randomized studies that concurrently assess various elements of IFN-based agents; and 2) limited research on biomarkers of IFN induction in viral respiratory infections. Our observations on the lack of randomized studies and need for further research agree with other recent reviews [21–23, 25].

First, few studies concurrently assessed different modes, dosages, frequencies, or duration of administration of IFN-based agents, making comparisons between these factors difficult given marked variations in study designs. Few studies assessed different dosages [41, 43, 45, 47, 53]. Additionally, no studies assessed different IFN molecular entities side-by-side. Thus, there is a need to understand optimal IFN type, mode of administration, dosage, frequency, and duration to balance safety, tolerability, and efficacy of these treatments in rigorously designed randomized trials. In terms of timing, there was limited research (i.e., only two studies) on the impact of IFNs or TLR-agonists as PEP [53, 54]. Additional research needs to be conducted in PEP to better understand the role that IFN-based agents may play in this setting, as control of infections post-exposure is critical for epidemic control.

Second, we also noted limited research on biomarkers of IFN induction, or downstream effects, or virologic endpoints in the scope of this review. Our review aimed to understand the roles between Type I and Type III IFN in response to viral infections in the respiratory tract, and what IFN signatures may exist to predict severe disease or response to IFN-targeted therapies in viral respiratory infections. There were no studies which corresponded to this research question, which we suspect is largely due to our exclusion of non-clinical studies. The lack of understanding of how IFN or IFN-inducers influence local or systemic IFN-stimulated gene responses is critical for more holistic understanding of the timing, dose, and route of administration for IFN-based agents' use in humans. Thus, we advocate for inclusion of biomarker data in clinical trials. RNA sequencing, multiplexed cytokine measures, and other highly dimensional modern assays can address this outstanding question of gene responses. For example, there were no studies focusing on IFN-induced gene signatures or investigating local responses (e.g., gene signatures or cytokine secretion), which could inform optimal dosing and timing of these therapies for viral respiratory infections. This would be important to address in future research given the need to understand target coverage across tissue types. Also, notably, most human challenge studies used clinical, rather than quantitative/viral shedding endpoints. This is an important limitation of these studies, as understanding viral shedding and downstream community viral spread is of utmost importance during a pandemic. Therefore, studies of innate immunity agents should employ frequent upper respiratory sampling for quantitative PCR/RT-PCR to determine if these agents are truly reducing viral replication. Lastly, previous work has not included study of pre-existing autoantibodies to IFN, which studies have shown to be associated with poor outcomes after COVID-19, or the effect of genetic variation in innate immunity genes, which could be associated with differential responses to IFN-pathway drugs [14]. Thus, future clinical studies need to be attuned to this phenomenon of antibody generation, as efficacy of the agent may be reduced.

While this is one of the first scoping reviews to examine the existing literature on IFN-based agents on viral respiratory infections, it has several limitations. First, many of the studies

included in our review were older (from before the 1990s), when IFN-based agents were first discovered, and knowledge of IFN-inducing agents was emerging. These older studies rarely included data on mechanisms of action or biomarkers, and measurements of efficacy and even identification of the IFN-agent used were difficult to ascertain. This lack of more recent data may have resulted from early studies demonstrating low efficacy as well a general decline in the use of human challenge studies. Additionally, a general shift in interest in the research community in small molecules and pathogen-specific agents may have driven disinterest in the broadly acting IFN-based agents. However, the number of IFN-related studies published and ongoing during the COVID-19 pandemic signal a renewed interest in these agents. Second, we limited our search to human clinical trials for viral respiratory infections and excluded human cell lines and animal studies, which may have provided important insights, particularly on biomarkers. Third, we may not have captured all agents that could induce or stimulate IFN, particularly if any newly discovered agents evaded our search terms. Fourth, we excluded studies that used IFN-based agents for non-respiratory infections, leading to exclusion of dozens of studies on multiple sclerosis and hepatitis or on the use of TLRs as adjuvants for vaccines. However, we do not believe these studies on chronic administration would provide relevant insight on the role of IFN-based agents in the context of time-limited administration for a respiratory virus epidemic. Additionally, although we re-ran the searches several times throughout the study, articles published after February 2021 are not included in this analysis given how quickly and continuously the literature is evolving for COVID-19. Lastly, our review included only literature published in English and studies with full text accessibility through a library service, which may have excluded some relevant studies. However, we sought out several library services to include as many articles as possible. Notwithstanding these limitations, our scoping review is timely and insightful as the COVID-19 pandemic evolves and planning for future epidemic preparedness is underway.

## Conclusions

Through a systematic mapping of the existing literature on the use of IFN-based agents, including TLR-agonists, for human viral respiratory infections, this scoping review provides insights on the existing literature and highlights key gaps, with an eye for epidemic preparedness for the future. IFN-based agents warrant further research, particularly on the administration routes, efficacy, and biomarkers, including IFN signatures, autoantibodies, and gene alterations that may impact patient response. Several studies investigating the use of IFN-based agents in viral respiratory infections, including for SARS-CoV-2, are ongoing, signaling the potentially promising role of IFN-based agents in mitigating future viral respiratory epidemics.

## Supporting information

**S1 Table. Preferred Reporting Items for Systematic reviews and Meta-Analyses extension for Scoping Reviews (PRISMA-ScR) checklist.**
(DOCX)

**S2 Table. Search terms.**
(DOCX)

**S3 Table. Study selection criteria.**
(DOCX)

**S4 Table. Data extracted from articles.**
(DOCX)

## Acknowledgments

We would like to thank Katie Newhall for graciously providing her expertise and support for this work. We would like to the acknowledge the support of the staff and leadership at the START Center.

## Author Contributions

**Conceptualization:** Aldina Mesic, David M. Koelle, Rena C. Patel.

**Data curation:** Emahlea K. Jackson, Mathias Lalika.

**Formal analysis:** Aldina Mesic, Emahlea K. Jackson, Mathias Lalika.

**Investigation:** Aldina Mesic.

**Methodology:** Aldina Mesic, Emahlea K. Jackson, Mathias Lalika, Rena C. Patel.

**Project administration:** Aldina Mesic, Rena C. Patel.

**Software:** Emahlea K. Jackson, Mathias Lalika.

**Supervision:** Rena C. Patel.

**Validation:** David M. Koelle.

**Writing – original draft:** Aldina Mesic, Rena C. Patel.

**Writing – review & editing:** Emahlea K. Jackson, Mathias Lalika, David M. Koelle, Rena C. Patel.

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
