## [Decision Letter · Decision Letter 0]

15 Dec 2021

PGPH-D-21-00739

Interferon-based agents for current and future viral respiratory infections: a scoping literature review of human interventional trials

Dear Aldina Mesic,

Thank you for submitting your manuscript to PLOS Global Public Health. After careful consideration, we feel that it has merit but does not fully meet PLOS Global Public Health’s publication criteria as it currently stands. Therefore, we invite you to submit a revised version of the manuscript that addresses the points raised during the review process.

We look forward to receiving your revised manuscript.

Kind regards,

Muhammad Asaduzzaman, MD MPH MPhil

Academic Editor

Journal Requirements:

1. Please provide separate figure files in .tif or .eps format only, and remove any figures embedded in your manuscript file.  If you are using LaTeX, you do not need to remove embedded figures.

2. Please amend your Data Availability Statement and indicate where the data may be found.

3. Please amend your detailed Financial Disclosure statement. This is published with the article, therefore should be completed in full sentences and contain the exact wording you wish to be published.

ii). State the initials, alongside each funding source, of each author to receive each grant.

Additional Editor Comments (if provided):

Reviewers' comments:

Reviewer's Responses to Questions

**Comments to the Author**

1. Does this manuscript meet PLOS Global Public Health’s publication criteria? Is the manuscript technically sound, and do the data support the conclusions? The manuscript must describe methodologically and ethically rigorous research with conclusions that are appropriately drawn based on the data presented.

Reviewer #1: Yes

Reviewer #2: Yes

2. Has the statistical analysis been performed appropriately and rigorously?

Reviewer #1: N/A

Reviewer #2: N/A

3. Have the authors made all data underlying the findings in their manuscript fully available (please refer to the Data Availability Statement at the start of the manuscript PDF file)?

Reviewer #1: Yes

Reviewer #2: No

4. Is the manuscript presented in an intelligible fashion and written in standard English?

Reviewer #1: Yes

Reviewer #2: Yes

5. Review Comments to the Author

Reviewer #1: I appreciate the effort of the authors for the interesting article. The article is well written, but there are some methodological disparities. There is scope for improvement. Please find my comments here:

1. As per the PRISMA-ScR checklist, the protocol should be registered. Did the author register the protocol?

2. Study Selection (Page 4): The title and abstract screening should be conducted by two independent review authors. It is not mentioned. Did the authors screen independently?

2. Study Selection (Page 4): The full text screening should be conducted by two independent review authors. It is not mentioned. Did the authors screen independently? If independent screening is not done, it should be mentioned in the limitation.

3. Ideally, the data extraction should be conducted by two independent review author. Please mention if not done so.

4. In inclusion criteria, the authors have mentioned that only peer reviewed articles were included. What was the logic for searching medRxiv? Is medRxiv a database for peer reviewed articles?

5. As per PRISMA-ScR checklist, quality assessment should be conducted. Authors might use the JBI quality assessment checklist for RCTs and other study designs.

Reviewer #2: Major comments:

Lines 349-350: “While this is one of the first scoping reviews to examine the existing literature on IFN-based agents on viral respiratory infections, it has several limitations.” Could you add the reference of the existing scoping reviews and also compare similarities and differences with your scoping review, which I guess is the most updated and it would be interesting to know, what your review adds to the existing ones.

There are many systematic reviews (SRs) addressing interventions for COVID-19, including interferon-based therapies. Do your results agree with these SRs?

The authors did not restrict the search to adults only. Where results stratified by age? Where there any studies on children?

Minor comments:

Literature search:

Lines 124-125: The authors conducted the searches in October and December 2020 and February 2021. Why did you skip the months of November 2020 and January 2021? Or did you mean from October 2020 to February 2021?

Line 128: The authors included human clinical trials. However, in the Supplementary Materials 3: Study selection criteria table they included also observational studies. Although in the Supplementary Materials 2: Search terms table they only added search terms for clinical trials and not for observational studies “(Clinical Trial[sb] OR Multicenter Study[sb] OR "Clinical Trials as Topic"[Mesh] OR "Multicenter Studies as Topic"[Mesh”)

In general, please clarify the target population. Are children and adults included? I do not know about scoping reviews but of systematic review, still I think it would be good to have the P of the PECOS.

Results:

Lines 158-164: Please check the first paragraph related to Figure 1. The numbers given do not quite tally. Please correct.

Figure 1: There are several mistakes listed here:

Abstracts screened: 977 and excluded 744. Thus, full text articles assessed for eligibility should be 233, or excluded should be 783 if the full articles assessed were truly 194…Please correct

Records excluded during full text review do not add up to 146 but to 159. Please correct.

Table 1: It would be good to also have the info on the country these studies were published, to have an idea where this topic is relevant, and also it will give us indirect information on ethnicity (efficacy of treatment may vary among different ethnic groups, maybe this explains variability in efficacy).

Tables in general: In those observational studies, how where participants recruited. Where there recruited from the general population, from hospitals etc.

Line 166: The authors found 16 studies published between 1981 and 1990. If the search was conducted in 2020-2021 one would expect to only retrieve those articles published in this period of time. I think what has happened here is a case of a misleading report: maybe the authors conducted a search from inception until February 2021 but did not report it appropriately? It is difficult to understand when you truly conducted the search. You may have conducted it initially from inception to October 2020, then updated it until December 2020 and then further expanded covering until February 2021. (lines 366-369 you finally explained what I understood… that you re-ran the searches several times. Still you have to make it clear what your search covers.)

Results are reported per type of IFN. Since there are no subheadings I think for the reader it would be great if the authors start with IFN-alpha they continue to use the same terminology. For example, in lines between 181 to 196 one can read: IFN-α, IFN, and IFN-a. It also applies to other types: IFN- β, IFN-B etc. Please keep one at a time. IFN when general, IFN-α when discussing results related to alpha and the same for the rest of the types. It is in general applicable to all sections within results.

Table 2: Regarding the study design, some are labelled as randomized and others have no information on randomization. Given that you only included RCTs and observational studies, either you add the term in the rest of the studies or you mention it more clearly in the inclusion criteria in the text (line 128). There is one study that is longitudinal, open-label. Is it randomized, or is it an observational study? It would be good to differentiate both type of studies. Could you standardize the information provided e.g.:

Randomized

Parallel or cross-over etc

Type of blinding (open label, single- or double-blind, or observer-blind)

Placebo-controlled

Challenge

Observational (type of design: cross-sectional, cohort, case-control etc)

Treatment Results:

Line 239: 7 studies used IFN-α. However only 6 references are provided in the text: 43, 45-49.

Line 266: Hung et al. has no reference number in the text.

Line 268: Monk et al has no reference number in the text.

Line 276: Put the one administering IFN-λ in a separate paragraph, otherwise you think it is a study administering IFN-κ. Just to avoid confusion and following the pattern you’ve been using throughout the manuscript.

Table 4: You have included non-randomized trials but according to the Supplementary Materials 3: Study selection criteria table you included randomized clinical trials. This deserves some clarification. What were the inclusion and exclusion criteria regarding the study type then?

In the results section it should be described how many studies were of clinical design and how many were of observational design since the strength of the evidence will be different. To assess efficacy and safety, clinical trials are recommended over observational studies.

Lines 305 and 311: Could you please add the references of the studies that commented on the pharmacokinetic and pharmacodynamic outcome data? And also indicate the type of IFN administered?

6. PLOS authors have the option to publish the peer review history of their article (what does this mean?). If published, this will include your full peer review and any attached files.

**Do you want your identity to be public for this peer review?** For information about this choice, including consent withdrawal, please see our Privacy Policy.

Reviewer #1: No

Reviewer #2: No

---

## [Editor Report · Decision Letter 1]

17 Feb 2022

Interferon-based agents for current and future viral respiratory infections: a scoping literature review of human studies

PGPH-D-21-00739R1

Dear Aldina Mesic,

We are pleased to inform you that your manuscript 'Interferon-based agents for current and future viral respiratory infections: a scoping literature review of human studies' has been provisionally accepted for publication in PLOS Global Public Health.

Best regards,

Muhammad Asaduzzaman, MD MPH MPhil

Academic Editor